# DISTILLING TEXT-IMAGE FOUNDATION MODELS

## ABSTRACT

Large pretrained foundation models (such as CLIP) are among the most recent significant advances in the AI community. Their implication is profound. This paper examines the value of these foundation models as a model knowledge base – we aim to distill the knowledge in these foundation models for training lightweight models designed for specific tasks in practical application scenarios with improved performance. Despite abundant progress in knowledge distillation (KD) in traditional models trained under the supervision of class labels in datasets encoded as integers, distilling such text-image contrastive learning model has not been explored extensively. Meanwhile, KD is well-known for being bothered by the capacity gap problem (i.e., distilling knowledge from a teacher significantly larger than a student often degrades the performance of the student). The teacher-student capacity gap in distilling foundation models is even larger. Therefore, how to overcome this potential issue is also elusive now. This paper presents detailed analyses of these questions aiming to successfully tap into a pretrained foundation model (CLIP) to boost the student's performance. Besides the practical performance benefits, several interesting discoveries are unveiled: (1) CLIP is not bothered by the capacity gap, which may let us re-evaluate if the "capacity-gap" issue is really due to the capacity gap (2) We find the reason is largely due to that CLIP is not over-confident on the wrong labels when misclassifies input image samples.

## 1 INTRODUCTION

Large, pretrained, foundation models (e.g., CLIP (Radford et al., 2021), DALL-E 2 (Ramesh et al., 2022) and GPT-3 (Brown et al., 2020)) are capable of many complex tasks such as zero-shot prediction - the ability of models to predict the classes to which the input samples belong during testing without previous exposure to samples from that classes during training, generating images according to text prompts, generating images inspired by their originals, translating, reading comprehension, etc. However, the scales, or the numbers of parameters that these models contain are so large that it would be difficult to deploy such models to devices with limited computing power such as mobile phones, tablets, and laptops. In addition, even though these foundation models are versatile, demonstrating great competence in abundant tasks that are considered to be challenging for regular neural networks, in some situations, however, instead of using all the functions that these models are capable of, we may only need to use parts of or even a derivative of their functions. These facts indicate that deploying a full foundation model in all use cases could be a waste of computational resource and memory, and such intentions could be even impractical in some situations. Therefore, the study of techniques that could be applied to compress or enable the utilization of a portion of the functions of such foundation models would be valuable and necessary.

To use a portion or derivatives of the functions of these huge, pretrained foundation models, one promising mechanism is to transfer the knowledge from the foundation models to lightweight, task-specific models. In (Hinton et al., 2014), a knowledge distillation (KD) algorithm is proposed, which is able to improve the task-specific performance of a model with a smaller scale (the student network) by transferring knowledge from another model with a larger scale and better performance specific to the task to it. The KD algorithm proposed by Hinton et al. (2014) (HKD) aims at minimizing both the Kullback-Leibler divergence (KL divergence) loss between the outputs of the teacher network and the student network along with the cross entropy loss between the student network and class labels. However, given the differences in network architectures along with pretraining methods between foundation models and conventional models, applying HKD directly on foundation models

may not be an effective approach to exploit the knowledge within the foundation models to benefit the performance of lightweight models designed for definite tasks. Firstly, the teacher network, in this case, a foundation model, is not pretrained to optimize its performance on the particular task that the student network is designed for. Moreover, the intrinsic properties of the dataset utilized for pretraining foundation models could be different from that of the dataset we adopt for a specific task. In addition, in (Cho & Hariharan, 2019; Mirzadeh et al., 2020), the existence of "capacity gap" between a teacher and a student is believed to be the major factor that prevents the performance of a student network from further improving when the teacher network contains more parameters and have better task-specific performance. When a foundation model, which contains a considerably larger quantity of parameters compared to conventional models, is adopted as a teacher network for knowledge distillation, this problem could become even more severe.

In this paper, we focus on the image classification task, exploring and investigating knowledge distillation-related properties of a pretrained foundation model CLIP (Radford et al., 2021) under various experimental settings.

Our contributions are:

- We notice that naively distilling knowledge from CLIP (Radford et al., 2021) to student networks does not lead to satisfactory results, meaning such student networks do not outperform those distilled from more commonly adopted teacher networks (e.g., ResNet 34, 50 (He et al. (2016))). We hence propose a process to improve the accuracy of the teacher network on image classification before knowledge distillation, which is the fine tuning of CLIP. This accuracy is the upper bound of that of the student network.

- We find that distilling from CLIP is not vulnerable to the "capacity gap" issue even when the difference in the number of parameters between the teacher network and the student network reaches more than a thousand times. Moreover, when there are only limited training samples available, the superiority of CLIP in knowledge distillation increases. Our experimental results suggest the reason may well be related to the training recipe of CLIP instead of the network architecture. Our further quantitative analysis of the output of teacher networks reveals that it is more probable for image classifying models trained with cross-entropy criterion to give a high score to a wrong label on misclassification, which can later mislead the student network in knowledge distillation. On the contrary, giving a relatively high score to wrong labels is less likely for models trained under CLIP paradigm. This can have a profound impact on the understanding of the capacity gap issue

- Based on these findings, we assign our finetuned CLIP to supervise the training of the lightweight model MobileNetV3 (Howard et al., 2019), a network designed for CPU deployments. The achieved performance turned out to be notably higher than that of those trained from scratch or under the supervision of regular networks.

## 2 RELATED WORK

**Knowledge distillation.** Buciluǎ et al. (2006); Hinton et al. (2014) proposed to improve the performance of lightweight models on particular tasks (e.g., image classification, speech recognition) by forcing such models (the students) to mimic cumbersome, over-parameterized models (the teachers) on the output level. Romero et al. (2015) followed this notion and proposed to maximize the similarity between the student and the teacher with respect to feature maps of hidden layers. Tian et al. (2020) proposed a contrastive learning objective, which allows a student network to learn much more important information from the data representation produced by a teacher network. In other works related to knowledge distillation, the knowledge to be transferred from a teacher to a student is defined as the association among input samples (Park et al., 2019; Tung & Mori, 2019), the probabilistic distributions of features (Passalis & Tefas, 2018), etc.

**Capacity gap issues.** Intuitively, with the supervision of a more complex teacher network comprising more parameters, the student network should be trained to perform better. In reality, however, the performance of a student network could not be enhanced indefinitely or become arbitrarily close to that of its corresponding teacher network. Cho & Hariharan (2019) pointed out the phenomenon that larger models may not correspond to better performing student networks, which was explained by their "mismatched capacity". They proposed to adopt *early-stopped* knowledge distillation (Cho

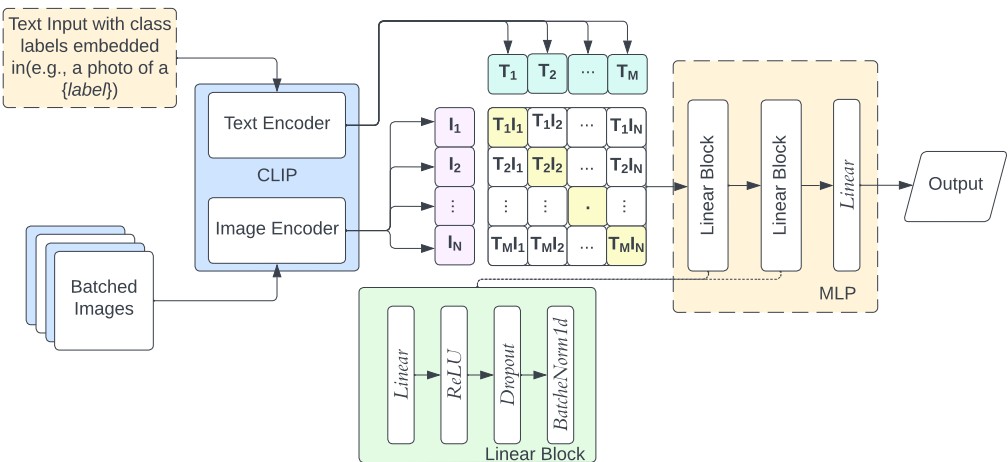

Figure 1: Finetuned CLIP system architecture. The standard prepossessing procedures, with data augmentation for ImageNet (Deng et al., 2009b) (e.g., random resized crop, horizontal flip, normalization, etc.) are applied to the batched images as the input of CLIP image encoder. Class labels in the dataset we adopt (*imagenet100*) are embedded in phrases as the input of CLIP text encoder (e.g., a photo of a {*husky*}). The labels are given extra context information to eliminate ambiguities and to explain proper nouns (e.g., *kuvatz → kuvatz, a type of dog*; *fig → fig, a type of fruit*). The pretrained CLIP gives positive text-image pairs high *cosine similarity* while suppressing that of the negative pairs, such that an image is matched to its corresponding label in the form of text. A multi-layer perceptron appended to CLIP (Radford et al., 2021) takes in its output. Parameters in the MLP are optimized in the finetuning process, during which the parameters in CLIP are frozen. The finetuning process is supervised by the integer-encoded labels in our *imagenet100* dataset.

& Hariharan, 2019) to improve the performance of the student networks. Mirzadeh et al. (2020) proposed to improve knowledge distillation performance by adopting intermediate-sized models to compensate for the capacity gap between the teacher networks and the ultimate student networks.

**Foundation models.** Foundation models are those with vast scale and trained on a large amount of data, such that they are competent in various downstream tasks (Bommasani et al., 2021). In (Brown et al., 2020), GPT-3, a model with 175 billion parameters, demonstrated its prominent ability in reading comprehension, commonsense reasoning, translating, etc. The "Bidirectional Encoder Representations from Transformers" or BERT (Kenton & Toutanova, 2019), a language model pretrained on unlabeled text can be adapted for a variety of natural language-related tasks (e.g., language inference, question answering) without major architectural modifications for specific tasks. Instead, finetuning the pretrained model with an extra output layer would be sufficient. CLIP (Radford et al., 2021) models trained on a dataset consisting of 400 million text-image pairs, are competent in the zero-shot task on multiple datasets. DALL-E 2 (Ramesh et al., 2022), a model that adopts the framework of CLIP (Radford et al., 2021), is able to generate images based on text input and variations of images inspired by the originals.

**Knowledge distillation on foundation models.** Knowledge from foundation models can be transferred to a variety of lightweight, task-specific models through knowledge distillation and hence improve their performance. In (Chen et al., 2019; Tang et al., 2019), knowledge within pretrained BERT (Kenton & Toutanova, 2019) was distilled to leverage small-scaled models designed for specific tasks such as natural language understanding, text generation, sentiment classification, etc. Notably, before knowledge distillation, the teacher network –BERT (Kenton & Toutanova, 2019) was finetuned. Jiao et al. (2019) managed to shrink BERT (Kenton & Toutanova, 2019) as a whole through knowledge distillation without damaging its versatility and performance on different tasks. Wang et al. (2022) proposed multimodal adaptive distillation to improve the performance of unimodal encoders in vision-language tasks (e.g., visual commonsense reasoning, visual question answering, visual entailment, etc.).

## 3 METHOD

### 3.1 PREREQUISITES: KNOWLEDGE DISTILLATION

In this paper, we adopt the knowledge distillation proposed by Hinton et al. (2014) as the technique to transfer knowledge from CLIP to lightweight models, and we refer to this algorithm as Hinton knowledge distillation (HKD). The complete objective of HKD is a linear combination of two sub-objectives:

$$\mathcal{L}_{\text{HKD}} = \alpha \mathcal{L}_{KLDiv} + \beta \mathcal{L}_{CE}, \tag{1}$$

where $\alpha$ and $\beta$ are adjustable hyper-parameters weighting the Kullback-Leibler (KL) divergence loss $\mathcal{L}_{KLDiv}$ and the cross-entropy loss $\mathcal{L}_{CE}$ respectively. The cross-entropy loss asks a student network to learn from the hard labels of datasets:

$$\mathcal{L}_{CE} = H(y, \boldsymbol{y}^{(s)}), \tag{2}$$

where $y$ denotes the class labels encoded as integers and $\boldsymbol{y}^s$ denotes the output of a student network. The KL divergence loss asks the student to mimic the teacher on the output level. In the calculation of KL divergence loss, a hyperparameter named *distillation temperature* $\tau$ is introduced to soften the output of both the teacher and the student, allowing the probability distribution of teacher output to be more informative:

$$\mathcal{L}_{KLDiv} = \tau^2 KL(\sigma(\boldsymbol{y}^{(t)}/\tau)|\sigma(\boldsymbol{y}^{(s)}/\tau)), \tag{3}$$

where $y^{(t)}$ denotes the teacher output and $\sigma$ denotes softmax function.

### 3.2 FINETUNING CLIP

CLIP (Radford et al., 2021) stands for contrastive language-image pretraining. The two major components of the model are an image encoder and a text encoder. The inputs of CLIP are text-image pairs and the pretraining of the model enables the image encoder and the text encoder to generate adequate representations of the input images and text respectively. In addition, the representation of an image is trained to match the corresponding text representation by maximizing the cosine similarity between positive pairs while minimizing that between negative pairs. That is, let $I_{1,...,i}$ be the normalized image features and $T_{1,...,j}$ be the normalized text features. The objective of pretraining is to maximize $I_i T_j^T$ for $i = j$ and minimize $I_i T_j^T$ for $i \neq j$.

The finetuning of CLIP aims at optimizing the system performance on specific tasks and datasets, which contains two processes: the improvement of text prompts and the refinement of model output. Figure 1 gives an illustration of the system architecture, in which CLIP (Radford et al., 2021) is implanted.

#### 3.2.1 TEXT PROMPTS IMPROVEMENT

For image classification task, the text input of CLIP is usually class names embedded in sentences or phrases (e.g., this is a photo of a {*class*}) and for CIFAR-10 dataset (Krizhevsky et al., 2009), the class names could be automobile, airplane, horse, etc. However, for larger datasets with more classes like ImageNet (Deng et al., 2009a), some class names become ambiguous due to polysemy while others could be proper nouns. Our approach is providing extra descriptions to or specifying the parent class of certain labels. For instance, we replace the labels *Model T*, which refers to a type of motor vehicle manufactured by FORD, with *Model T, automobile, car*, and substitute *Saint Bernard* with *Saint Bernard, a type of dog* respectively according to what the labels in the dataset actually refer to.

#### 3.2.2 OUTPUT REFINEMENT

To improve the performance of CLIP on a specific task and dataset, the output of CLIP is refined. The procedures for refining the output of CLIP are: (1) appending extra multilayer perceptrons (MLP) fitting a particular dataset for image classification to the CLIP model; (2) freezing all the parameters in CLIP; (3) optimizing the parameters in the MLP on a dataset under the supervision of integer-encoded labels using conventional cross entropy criterion on the image classification task. This process can be mathematically formulated as below.

Assume $\boldsymbol{x}, \boldsymbol{t}$ to be the image and text input to CLIP respectively, and $\boldsymbol{w}_{\text{CLIP}}$ to be the parameters in CLIP. Then we denote the output of CLIP to be:

$$\boldsymbol{y} = f(\boldsymbol{w}_{\text{CLIP}}, \boldsymbol{x}, \boldsymbol{t}). \tag{4}$$

The contrastive output of CLIP $\boldsymbol{y}$, in which an image embedding is matched to its corresponding text embedding, will be passed to the input layer of the adjoining MLP. Let $\boldsymbol{w}_{\text{MLP}}$ be the parameters in the MLP, and hence its output can be written as:

$$\boldsymbol{z} = h(\boldsymbol{w}_{\text{MLP}}, \boldsymbol{y}). \tag{5}$$

Let $\boldsymbol{z}'$ represents the ground-truth vector of labels encoded as integers, the objective function of output refinement can then be expressed as:

$$\mathcal{L}_{\text{refine}} = H(\boldsymbol{z}', \boldsymbol{z}), \tag{6}$$

which is to be minimized with respect to $\boldsymbol{w}_{\text{MLP}}$ through training.

### 3.3 Exploring the Capacity Gap

Capacity gap or *mismatched capacity* (Cho & Hariharan, 2019) is considered to be a significant factor leading to the phenomenon that a teacher with higher capability may not necessarily further enhance the performance of a given student network in knowledge distillation. In this part, we propose our approaches to examine the capacity gap resistance property of CLIP and a metric revealing the reason explaining why distilling CLIP is not bothered by the capacity gap.

#### 3.3.1 Examine Capacity Gap Resistance Property

**Baseline comparison.** Under this setting, a common, relatively small-scaled convolutional neural network (CNN) is selected to be the student, while regular CNNs and the finetuned CLIP model are selected to be the candidate teacher networks. Comparisons are conducted among the accuracy of student networks with identical structures but distilled from different teacher networks. The involvement of regular CNN with different scales is intended to demonstrate the negative impact of the capacity gap.

**Reduced student network width.** Reducing the widths of student networks (CNN) means decreasing the number of filters in each of the convolution layers, resulting in a reduction in the number of parameters in student networks. With the candidate teachers unchanged, the difference in parameter number or the gap in capacity within teacher-student pairs will be enlarged, and hence the capacity gap resistance of the teacher networks can be further justified.

**Low-shot classification.** In this case, the students and teachers are both exposed to a limited quantity of training images. Specifically, given a dataset $D$, the train set of $D$ is denoted as $D_{train}$. For each class in $D$, $k$ pictures in $D_{train}$ is selected to form the train set for low-shot classification, while the test set of $D$ denoted as $D_{test}$ is adopted directly without any modification. This setting investigates the influence of the capacity gap under the situation of low training samples.

#### 3.3.2 Capacity Gap Resistance Related Metric

In knowledge distillation, the media allowing the knowledge to be transferred from the teacher and the student is their output. The resistance to the capacity gap should be related to one or more quantifiable features within the output of the teacher networks. In the image classification task, the model output corresponding to an input image is an $n$-dimensional vector $\boldsymbol{y}^o$, where $n$ is the total number of classes in the given dataset. A well-trained model would assign the highest score to the element in $\boldsymbol{y}^o$ matching the class to which the input image belongs, otherwise, the input image is deemed to be mistakenly classified. That is for $label \in \{0, ...i, ..., n-1\}$ and $\boldsymbol{y}^o = [y_0^o, ..., y_i^o, ..., y_{n-1}^o]$, where $i \in [0, n-1]$. Assume an input image is with label $i$, and the image is considered to be correctly classified if and only if $y_i^o$ is the maximum element in $\boldsymbol{y}^o$. We believe that when the teacher misclassifies an input image, meaning the highest score is assigned to the element in $\boldsymbol{y}^o$ not matching the label of the input image, if the score is relatively high (*over confidence*), the student could hence be misguided. Therefore, we propose a probabilistic metric to evaluate the above-mentioned phenomenon that occurs in teacher network output:

$$p = \frac{N_{err\&oc}}{N_{err}}, \tag{7}$$

Table 1: Student Accuracy (%) on test set of *imagenet100* among different teacher-student pairs. ResNet 18 (He et al., 2016) is adopted as the student network. "Params" stands for the number of parameters in models and it is calculated in millions. "Param Gap" denotes the gap in parameter number between the teacher and student, which is measured in the number of times. "Scratch/None" stands for training from scratch without the supervision of teacher networks.

| ResNet 18 + ImageNet100 | | | | |
|---|---|---|---|---|
| Teacher network | Teacher Accuracy(%) | Student Accuracy(%) | Params | Param. Gap |
| Scratch /None | / | 84.10 | 11.22 M | 1.00 $\times$ |
| ResNet 34 (He et al., 2016) | 85.98 | 85.88 | 21.32 M | 1.90 $\times$ |
| ResNet 101 (He et al., 2016) | 87.36 | 85.80 | 42.60 M | 3.80 $\times$ |
| Raw CLIP (Radford et al., 2021) | 90.77 | 85.78 | 291.00 M | 26.05 $\times$ |
| Finetuned CLIP (ours) | 95.88 | **86.02** | 291.27 M | 26.07 $\times$ |

where $N_{err}$ represents the number of misclassified samples and $N_{err\&oc}$ represents the number of samples that are both misclassified and the corresponding output vectors experience *over confidence*. In this paper, we define an output vector $\boldsymbol{y}^o$ of a model is *over confidence* if:

$$max_{i\in[0,n-1]}y_i^\sigma >= \gamma, \tag{8}$$

where $\boldsymbol{y}^\sigma = \text{softmax}(\boldsymbol{y}^o)$, $\boldsymbol{y}^\sigma = [y_0^\sigma, ..., y_i^\sigma, ..., y_{n-1}^\sigma]$, and $\gamma$ is an adjustable parameter.

# 4 EXPERIMENTS

## 4.1 BASIC SETTINGS

Experiments in this work are conducted on a machine with `4` NVIDIA GTX TITAN Xp GPUs. Data parallel technique is utilized. Dataset selection, networks involved in experiments, and configuration of hyperparameters are introduced as follows.

**Dataset** In this paper, we use a subset of ImageNet (Deng et al., 2009a) containing 100 classes randomly sampled from the original ImageNet dataset, and we call this dataset *imagenet100*. This dataset contains 1.2 million training samples and 50k testing samples, suggesting there are 1200 training images and 50 testing images per class. The sampling work only reduces the total number of classes and images compared to the original dataset, while the scales, aspects, and contents of the images remain unchanged.

**Networks** The networks involved in our experiments are ResNet 18, 34, 50, 101 (He et al., 2016), and CLIP (Radford et al., 2021). We choose pretrained ViT-L/14, a version of Vision Transformer (Dosovitskiy et al., 2021) to be the vision encoder in CLIP. The MLP appended to CLIP consists of trivial layers (see Figure 1) in neural networks: fully connected layers, batch normalization layers, dropout layers and we use ReLU as the activation function.

**Hyperparameter settings** We adopted part of the settings in (Matsubara, 2021), in which a slightly higher student accuracy was reported (71.37%) compared to that reported in (Hinton et al., 2014) (70.66%). Modifications have been made to the hyperparameter configuration to enable it to be suitable for our hardware. The number of training epochs in pretraining or finetuning teacher networks and knowledge distillation is set to `100`, the batch size is `128`. The initial learning rate is set to `0.1`, with a multi-step learning rate decay schedule at the epoch `60` and `90` by a factor of `0.1`. Stochastic gradient descent optimizer is chosen in our experiments, with a momentum of `0.9` and a weight decay of `1e-4`. In knowledge distillation experiments, we assign the distillation temperature $\tau$ to be `1`, indicating that no label softening is applied. The cross-entropy loss (weighted by $\beta$) between student output and class labels with integer encoding and the KL divergence loss (weighted by $\alpha$) between the output of teacher and student contribute equally to the total loss in knowledge distillation. That is, $\alpha = \beta = 0.5$.

## 4.2 BASELINE PERFORMANCE COMPARISON

We adopt ResNet 18 (He et al., 2016) to be the student network, ResNet 34, 101, CLIP (Radford et al., 2021) without being finetuned (Raw CLIP) and finetuned CLIP to be the candidate teacher

Table 2: Accuracy (%) comparison on test set of *imagenet100* among different teacher-student pairs, with reduced student network width. ResNet 18 (He et al., 2016) is adopted as the student network. "$(1/8)$" suggests that the number of filters in each convolution layer in residual blocks of the student network has reduced to $1/8$ compared to that in the original structure. "Params" stands for the number of parameters in models and it is calculated in millions. "Param. Gap" denotes the gap in parameter number between the teacher and student, which is measured in the number of times. "Scratch/None" stands for training from scratch without the supervision of teacher networks.

| **ResNet 18 $(1/8)$ + ImageNet100** | | | | |
|---|---|---|---|---|
| Teacher network | Teacher Accuracy(%) | Student Accuracy(%) | Params | Param. Gap |
| Scratch /None | / | 66.32 | 0.19 M | 1.00 × |
| ResNet 34 (He et al., 2016) | 85.98 | 66.48 | 21.32 M | 109.45 × |
| ResNet 101 (He et al., 2016) | 87.36 | 65.80 | 42.60 M | 218.70 × |
| Finetuned CLIP (ours) | 95.88 | **66.74** | 291.27 M | 1495.54 × |

networks. Except for raw CLIP, all models are pretrained or finetuned on our *imagenet100* dataset. From the results shown in Table 1, we observe that even when the difference in parameter number between the finetuned CLIP and ResNet 18 reached 26 times, the student still achieves the highest accuracy. In comparison, the parameter number difference between ResNet 101 and ResNet 18 is only 3.8 times but the student accuracy is slightly lower than that distilled from ResNet 34, which is a sign that the ResNet 101 - ResNet 18 pair is negatively influenced by the capacity gap in knowledge distillation while the finetuned CLIP - ResNet 18 pair is not. Further experiments are conducted to justify this observation.

## 4.3 REDUCED STUDENT NETWORK WIDTH

We further examine the observation we have in Section 4.2 that despite finetuned CLIP having the largest amount of parameters among all candidate teacher networks, knowledge distillation from it shows no sign of being influenced by the capacity gap. We enlarge the parameter number difference in a teacher-student pair by fixing candidate teacher networks while shrinking the number of filters in convolution layers in residual blocks in ResNet 18 (He et al., 2016), which is the student network in our work. In our experiment, the filter number in the student network is reduced to $1/8$ when compared to that in the original structure. Results in table 2 show that even the parameter number gap between finetuned CLIP and the student network becomes around 1.5 thousand times, the student network trained under the supervision of it maintains the highest accuracy among all students in different teacher-student pairs. In contrast, in the ResNet 101-ResNet 18

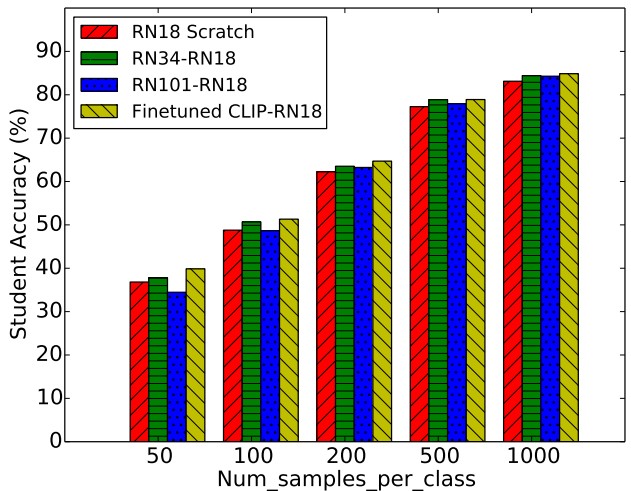

Figure 2: Low-shot distillation. Accuracy of student networks trained with the supervision of different teachers and exposed to different numbers of training samples in *imagenet100*. We choose candidate sample numbers in each class to be {50, 100, 200, 500, 1000}.

pair, even if there is a smaller gap in parameter number than that in finetuned CLIP-ResNet 18 pair, the student accuracy falls even lower than that trained from scratch. This could be viewed as a signal indicating that the capacity gap has a detrimental impact on knowledge distillation in ResNet 101-ResNet 18 pair.

Table 3: Low-shot distillation, where both students and teachers are exposed to a limited number of training samples. Accuracy (%) comparison on test set of *imagenet100* among different teacher-student pairs. ResNet 18 (He et al., 2016) is adopted as the student network. "Scratch/None" stands for training from scratch without the supervision of teacher networks. "Param Gap" denotes the gap in parameter number between the teacher and student, which is measured in the number of times. "$k$" denotes the number of training samples per class.

| ResNet 18 + ImageNet100 | | | |
|---|---|---|---|
| Teacher network | Student Accuracy (%) | Param. Gap | $k$ |
| Scratch /None | 36.82 | 1.00 $\times$ | 50 |
| ResNet 34 (He et al., 2016) | 37.80 | 1.90 $\times$ | 50 |
| ResNet 101 (He et al., 2016) | 34.46 | 3.80 $\times$ | 50 |
| Finetuned CLIP (ours) | **39.86** | 26.07 $\times$ | 50 |
| Scratch /None | 48.78 | 1.00 $\times$ | 100 |
| ResNet 34 (He et al., 2016) | 50.68 | 1.90 $\times$ | 100 |
| ResNet 101 (He et al., 2016) | 48.64 | 3.80 $\times$ | 100 |
| Finetuned CLIP (ours) | **51.32** | 26.07 $\times$ | 100 |
| Scratch /None | 62.24 | 1.00 $\times$ | 200 |
| ResNet 34 (He et al., 2016) | 63.52 | 1.90 $\times$ | 200 |
| ResNet 101 (He et al., 2016) | 63.24 | 3.80 $\times$ | 200 |
| Finetuned CLIP (ours) | **64.70** | 26.07 $\times$ | 200 |
| Scratch /None | 77.26 | 1.00 $\times$ | 500 |
| ResNet 34 (He et al., 2016) | 78.86 | 1.90 $\times$ | 500 |
| ResNet 101 (He et al., 2016) | 77.92 | 3.80 $\times$ | 500 |
| Finetuned CLIP (ours) | **78.90** | 26.07 $\times$ | 500 |
| Scratch /None | 83.12 | 1.00 $\times$ | 1000 |
| ResNet 34 (He et al., 2016) | 84.40 | 1.90 $\times$ | 1000 |
| ResNet 101 (He et al., 2016) | 84.28 | 3.80 $\times$ | 1000 |
| Finetuned CLIP (ours) | **84.86** | 26.07 $\times$ | 1000 |

## 4.4 LOW-SHOT CLASSIFICATION

We explore the capacity gap resistance under low-shot settings, meaning only a limited number of training samples are available. Similar to Section 4.2, candidate teachers are ResNet 34, 101, and Finetuned CLIP, with ResNet 18 to be the student network. Training data is a portion of our *imagenet100* dataset, that is, for each of the class in *imagenet100*, $k$ images in the original training set is sampled to form the training set for low-shot classification, where $k \in \{50, 100, 200, 500, 1000\}$. Each teacher is trained or finetuned on the low-shot training sets and later utilized to supervise the training of the student network. In other words, regular candidate teacher networks (ResNets), the MLP appended to CLIP and the student network is exposed to the same training set for low-shot classification in one low-shot setting. The results of low-shot classification are shown in Table 3 and Figure 2. We observe that with a lower average quantity of training samples with respect to the number of classes in the dataset, regular teacher networks become more vulnerable to the capacity gap, meaning the student accuracy degrades rapidly (ResNet 101 - ResNet 18). In contrast, for finetuned CLIP, the student network distilled from it consistently outperforms the rest especially when the number of available training samples is limited.

## 4.5 CAPACITY GAP RESISTANCE RELATED METRIC

In the work of CLIP (Radford et al., 2021), several vision models (image encoder) are trained including modified ResNet 50 and ResNet 101 (He et al., 2016) under the supervision of text with class labels embedded in, where the modifications are imposed on convolution layers and pooling layers outside residual blocks. We adapt these two pretrained image encoders to fit knowledge distillation on the image classification task. In addition, two networks having the same structure as the modified ResNet 50, 101 are trained respectively but under the regular cross-entropy paradigm supervised by integer-encoded labels of the dataset. To further investigate the capacity gap resistance property

Table 4: Comparing models with different pretraining methods, CLIP (Radford et al., 2021) versus regular cross-entropy paradigm. "*-img-enc" denotes pretrained image encoder in CLIP(Radford et al., 2021), "-mod-CE" denotes models having the same structures as that of the corresponding image encoders mentioned above but trained with cross-entropy loss. we again utilize ResNet 18 (He et al., 2016) with a reduced filter number in residual blocks as the student. $p$ is the metric proposed in Section 3.3.2, which measures the probability that the model gives a *relatively high* score to the wrong label in the model output on the condition that the highest score is assigned to a wrong label (misclassification). $\gamma$ is the threshold for determining whether a score is *relatively high*, where the scores are elements of a model output vector passed through the softmax function. In our experiment, the threshold is set to 0.5.

| ResNet 18 $(1/8)$ + ImageNet100 | | |
|---|---|---|
| Teacher network | $p(\gamma = 0.5)$ | Student Accuracy(%) |
| RN50-img-enc (Radford et al., 2021) | 0.29 | 65.70 |
| RN101-img-enc (Radford et al., 2021) | 0.31 | **66.66** |
| RN50-mod-CE | 0.41 | 65.86 |
| RN101-mod-CE | 0.39 | 65.44 |

of CLIP, the above-mentioned four models are assigned to be the candidate teacher networks and the ResNet 18 with reduced width is adopted as the student network in the following knowledge distillation experiment (see Table 4). For a model trained under the regular cross entropy paradigm, when compared to a model adopting the pretraining method of CLIP, it is more likely that it would give the wrong label a high score when misclassifying a sample, and we deem this *overconfidence* will misguide a student network in knowledge distillation.

## 4.6 EXTRA KNOWLEDGE DISTILLATION EXPERIMENTS

We extend the superior performance of CLIP in knowledge distillation to supervise the training of MobileNetV3 (Howard et al., 2019) and perform a series of knowledge distillation experiments on our *imagenet100* dataset. See Table 5.

Table 5: Accuracy (%) comparison on test set of *imagenet100* among different teacher-student pairs. MobileNetV3-L (Howard et al., 2019) is adopted as the student network. "Params" stands for the number of parameters in models and it is calculated in millions. "Param Gap" denotes the gap in parameter number between the teacher and student, which is measured in the number of times. "Scratch/None" stands for training from scratch without the supervision of teacher networks.

| MobileNetV3-L + ImageNet100 | | | |
|---|---|---|---|
| Teacher network | Student Accuracy(%) | Params | Param. Gap |
| Scratch /None | 80.94 | 2.77 M | 1.00 × |
| ResNet 34 (He et al., 2016) | 82.54 | 21.32 M | 7.68 × |
| ResNet 101 (He et al., 2016) | 82.00 | 42.60 M | 31.77 × |
| Finetuned CLIP (ours) | **84.76** | 291.27 M | 104.99 × |

## 5 CONCLUSION

In this paper, we have excessively examined that CLIP is robust to the impact of capacity gap issues in knowledge distillation under different experimental settings (extra small student network, low available training samples). We have demonstrated that the pretraining method of CLIP allows the model to overcome capacity gap issues because it is less likely for the model to be overconfident on the wrong class label when it misclassifies an input sample, which could mislead a student network during knowledge distillation. This encouraging result suggests that the knowledge within CLIP could be further exploited through knowledge distillation to benefit networks with even smaller scales designed to be deployed on devices with budget computational resources like mobile phones or those designed for tasks other than image classification.

ETHICS STATEMENT

To the best of our knowledge, the methods we proposed in this work and the experiments we conducted do not pose potential negative impacts on society and they comply with the ethical research standards of ICLR.

REPRODUCIBILITY STATEMENT

Code related to this work will be released upon publication for reproducing purposes.

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
