# OpenReview forum: "Distilling Text-Image Foundation Models"
_ICLR.cc/2023/Conference — Submitted to ICLR 2023_

### Official Review · Reviewer_JKPp · 2022-10-25

**Confidence:** 4
**Correctness:** 2
**Technical Novelty And Significance:** 2
**Empirical Novelty And Significance:** 2
**Recommendation:** 3

**Clarity, Quality, Novelty And Reproducibility:**

+ The writing is clear, the whole paper is easy to understand.
- The reproducibility of this paper depends on the release of the ImageNet-100 dataset the authors construct.
- Some redundancy in organization: content of Figure 2 duplicates with Table 3.

**Strength And Weaknesses:**

Strengths:
+ The finding that a finetuned CLIP does not suffer from "capacity gap" is interesting.
+ The finetuning method of CLIP seem novel in a limited degree.
+ Introducing a metric that evaluates the over-confidence of teacher models.

Weakness:
- The study exhibited shows limited novelty, which I consider insufficient for an ICLR submission.
  * Finetuning & distillation is a common practice for using large pre-trained models, for example the Big SimCLR [2].
  * Finetuning a CLIP using both its image encoder and text encoder has been used by previous literatures, for example VLKD [1].
  * Addressing the gap between teacher and student during distillation is indeed an import issue, as also mentioned by existing literatures such as [3], however, the scope of this paper is too limited (using CLIP as a teacher in image classification) to illustrate more values.
- Methods are evaluated only on one customized dataset, resulting in suspicious effectiveness under more experimental settings.
  * It is weird why the authors choose a customized subset of ImageNet (ImageNet-100) for evaluation instead of using the original ImageNet;
  * More public datasets such as CIFAR or others should be considered to prove that the effectiveness of the proposed method is solid.
  * Comparison of pre-trained teachers involves too few teacher types in Table 4, more should be included covering more network structures (e.g. ViT) and more pre-training strategies (e.g. MAE, MoCo, ...). Note that most these pre-trained backbones are publicly avaiable.
  * The relation between the threshold p and student accuracy is not extensively studied.

[1] Enabling Multimodal Generation on CLIP via Vision-Language Knowledge Distillation. ACL 2022.
[2] Big Self-Supervised Models are Strong Semi-Supervised Learners. NIPS 2020.
[3] Analyzing the Confidentiality of Undistillable Teachers in Knowledge Distillation. NIPS 2021.


**Summary Of The Paper:**

This paper studies the knowledge distillation performance on image classification when using a finetuned CLIP model as a teacher.
The authors find that distilling from a finetuned CLIP does not suffer from the "capacity gap" issue that commonly happens when teacher and student have a large difference. The CLIP-based distillation also have good performance when downstream training samples are limited. They also present an analysis and find that the strength of CLIP-base distillation is largely due to that CLIP is not over-confidence on the wrong labels when misclassifies input samples.

**Summary Of The Review:**

The study of this paper reveals that a finetuned CLIP can be served as a good teacher that does not suffer from "capacity gap".
This is an intersting direction that worth deeper investigation, for example, how could CLIP achieve such capability? Do other pre-trained models Does this conclusion holds for other datasets or tasks other than image classification?

However, this paper fails to conduct an investigation thorough enough on any valuable direction. The experiments are also too limited to support the claims of this paper w.r.t benchmarks, teacher models and hyper-parameters.

---

### Official Review · Reviewer_zsBj · 2022-10-26

**Confidence:** 4
**Correctness:** 2
**Technical Novelty And Significance:** 2
**Empirical Novelty And Significance:** 2
**Recommendation:** 3

**Clarity, Quality, Novelty And Reproducibility:**

Overall the experiments do not fully support the claims and hypotheses made in this paper. See Weaknesses for more details.

**Strength And Weaknesses:**

## Strength
1. this paper is well-written and is easy to follow.
1. The proposed method indeed improves knowledge distillation results, not by a large margin though.
1. the experiments clearly demonstrate that CLIP indeed suffers less from the capacity gap issue in knowledge distillation.

## Weakness
1. $\rho$ doesn't seem to be a reliable metric to determine whether a model performs better in knowledge distillation. Specifically, in Table 4, larger models trained with CE have smaller $\rho$. Since larger teacher models also achieve higher accuracy, then why larger teacher models achieve worse performance? Also, why does the authors choose ResNet type of CLIP model in Table 4 rather than ViT type of models as in other experiments?
1. It is hard to draw the conclusion that knowledge distillation from CLIP is superior to knowledge distillation from other pre-trained teachers as CLIP's accuracy is much higher than other teachers. It would be good to find a setting where the teachers perform roughly the same so that each model can be fairly compared. For example, the authors can use weaker/smaller CLIP models or evaluate on datasets where each model performs roughly the same.
1. Given that CLIP teachers achieve much higher acc, the performance improvement of the student model using CLIP is only marginal.
1. The experiments are not thorough enough. For example, the authors only demonstrate results on ImageNet-100. The authors only demonstrate results on ResNet type of student model. It would be good to also show ViT type of student model. It would be good to also show the results using other ViT type of teachers, not just CLIP.
1. the ablation of the improved text prompt is missing

**Summary Of The Paper:**

This paper focuses on improving knowledge distillation from CLIP to a much smaller network. The authors propose to enhance the CLIP teacher model by (1) text prompts improvement, and (2) output refinement. They also propose a metric $\rho$ to estimate how likely the teacher generates over-confident wrong supervisions.

**Summary Of The Review:**

Overall, the reviewer thinks the experiments are not thorough enough. The comparison between CLIP teacher and other ResNet type of teacher is not fair. $\rho$ does not seem to be a reliable metric.

---

### Official Review · Reviewer_wF3p · 2022-10-26

**Confidence:** 4
**Correctness:** 2
**Technical Novelty And Significance:** 2
**Empirical Novelty And Significance:** 2
**Recommendation:** 3

**Clarity, Quality, Novelty And Reproducibility:**

The quality and clarity of the writing is fair, although I think the paper can use more detailed experiments to improve the distillation performance and strengthen the comparison with existing works. Reproducibility seems reasonable. It’d be nice if the authors can promise to release the code.


**Strength And Weaknesses:**

Strength:
* The problem of distilling knowledge from large foundation models is well-motivated.
* The experimental findings of resistance to capacity gap are interesting.

Weakness:
* Resistance to the capacity gap is not very convincing. The gains of student models are very small over the baselines in Table 1 and 2. In my opinion, the proof of overcoming the capacity gap is to narrow or at least maintain the performance gap with the largest teacher. Table 1 shows that the gap widens with the strongest teacher model.
* The hypothesis that the resistance to capacity gap comes from less over-confident errors can use more study, e.g. what happens if we make the predictions less confident through adjusting the temperature parameter.
* Results in Table 5 show a bigger gap between CLIP model and other baselines. Any explanation for that?
* Tables 1-5 are all ablations and analysis. It’d be good to see some comparison with existing methods e.g. ESKD + AT (Cho et al. 2019).


**Summary Of The Paper:**

This paper presents analysis and empirical findings of distilling knowledge from large image-text foundational models (e.g. CLIP). The key observations  are 1) resistance to the capacity gap of the teacher/student models, 2) the image-text models are not overconfident on the wrong labels.

**Summary Of The Review:**

Although the problem is important and the initial results are interesting, the gains over the baselines are not significant enough. I’m also not convinced by the claims about overcoming the capacity gap (see weakness). Comparison with existing methods in experiments would also position this work better in the literature.

---

### Official Review · Reviewer_Af7J · 2022-10-27

**Confidence:** 4
**Correctness:** 2
**Technical Novelty And Significance:** 1
**Empirical Novelty And Significance:** 2
**Recommendation:** 3

**Clarity, Quality, Novelty And Reproducibility:**

Paper writing:
- Table 3 is not needed, since the information is already presented in Figure 2. I would suggest moving Table 3 to the appendix.
- Low-shot classification setting needs to be better motivated.
- Why are Table 4 experiments performed in the RN18 1/8 setting instead of normal RN18?

Code is not provided, thus this work is not reproducible.

**Strength And Weaknesses:**

Strengths:
- The experiments that are presented seem to be well executed.
- The proposed method for knowledge distillation from CLIP, using both image and text features, seems to be novel (though the standard baseline wasn't compared here).

Major Weaknesses:
- Why is the standard CLIP finetuning procedure not compared against? It is unclear as a reviewer whether the phenomena observed in the work is a result of the new proposed CLIP finetuning procedure or the standard CLIP finetuning procedure (which is widely used). In the standard procedure, the text encoder is thrown away, and a linear classification layer is appended to the end of the image encoder, which is then finetuned via the CE loss.
- The gap resistance analysis is severely underdeveloped and has a major flaw. In Table 4, is it not the case that the img-enc versions have higher accuracies than the mod-CE versions? Thus, the higher student accuracy could simply be explained by higher teacher accuracy. In order for the overconfidence ratio to hold weight, the accuracy confounder needs to be accounted for. For example, one way to do this would be to recalibrate the non-CLIP teacher model after training, which would lower the overconfidence ratio while keeping accuracy the same. If the proposed explanation is true, then this means the student network accuracy should increase.
- Limited experimental settings. It is unclear how many of the claims made in the paper are specific to ImageNet100 and the ResNet model family since the experiments are only conducted in this setting. To show that the phenomena is broader, more classification datasets are needed (Cifar10, cifar100, full ImageNet at the very least) and more model families are needed (ResNet, EfficientNet, ViTs, etc).

Minor weaknesses:
- All tables and figures should have error bars displaying stddev over random seeds.

**Summary Of The Paper:**

This paper applies knowledge distillation to CLIP for image classification and finds that student performance increases when using CLIP as a teacher despite the teacher being a larger size. Some initial explanatory answers are proposed.

**Summary Of The Review:**

In summary, this work evaluates a popular technique (finetuning CLIP) in a non-standard setting, and only on one dataset. An explanation is proposed for the phenomena, but this analysis is severely underdeveloped. In order to make this work ready for publication, proper baselines need to be implemented, the experiments need to be run on more classification datasets, and the analysis needs more experiments to control for confounders as well. These are all significant and major changes, hence my recommendation.

---

### Decision · Program_Chairs · 2023-01-20

**Decision:**

Reject

**Justification For Why Not Higher Score:**

I don't think the paper should be accepted.

**Justification For Why Not Lower Score:**

N/A

**Metareview: Summary, Strengths And Weaknesses:**

There is a consensus among reviewers that the paper, while addressing an interesting problem (knowledge distillation in CLIP like models) is not ready for publication. The shared concerns include, above all, unconvincing evaluation, both because it's done on a single dataset and because of the explanation for the reported phenomena (however interesting) being somewhat "underdeveloped". I agree.